# Erythrocyte sodium buffering capacity status correlates with self-reported salt intake in a population from Livingstone, Zambia

Sepiso K. Masenga[1]*, Leta Pilic[2], Malani Malumani[1], Benson M. Hamooya[1]

1 HAND Research Group, School of Medicine and Health Sciences, Mulungushi University, Livingstone, Zambia, 2 Faculty of Sport, Allied Health and Performance Science, St. Mary's University, Twickenham, London, United Kingdom

* sepisomasenga@gmail.com, smasenga@mu.ac.zm

**Data Availability Statement:** All relevant data are within the manuscript and its Supporting Information files.

## Abstract

### Background

Salt impairs endothelial function and increases arterial stiffness independent of blood pressure. The mechanisms are unknown. Recent evidence suggests that there is a possible link between salt consumption and sodium buffering capacity and cardiovascular disease but there is limited evidence in the populations living in Sub-Saharan Africa. The aim of our study was to explore the relationship between erythrocyte sodium buffering capacity and sociodemographic, clinical factors, and self-reported salt consumption at Livingstone Central Hospital.

### Methods

We conducted a cross sectional study at Livingstone Central hospital among 242 volunteers accessing routine medical checkups. Sociodemographic and dietary characteristics were obtained along with clinical measurements to evaluate their health status. Sodium buffering capacity was estimated by erythrocyte sodium sensitivity (ESS) test. We used descriptive and inferential statistics to describe and examine associations between erythrocyte sodium sensitivity and independent variables.

### Results

The median age (interquartile range) of the study sample was 27 (22, 42) years. 54% (n = 202) and 46% (n = 169) were males and females, respectively. The majority (n = 150, 62%) had an ESS of >120%. High salt intake correlated positively with ESS or negatively with vascular sodium buffering capacity.

### Conclusions

Self-reported high salt intake was associated with poor vascular sodium buffering capacity or high ESS in the majority of middle-aged Zambians living in Livingstone. The poor vascular sodium buffering capacity implies a damaged vascular glycocalyx which may potentially lead to a leakage of sodium into the interstitium. This alone is a risk factor for the future

**Funding:** This study received support from the Mulungushi University Research fund, awarded to SKM. The funders had no role in study design, data collection and analysis, decision to publish, or preparation of the manuscript.

**Competing interests:** The authors have declared that no competing interests exist.

development of hypertension and cardiovascular disease. However, future studies need to validate vascular function status when using ESS testing by including established vascular function assessments to determine its pathophysiological and clinical implications.

## Background

High intake of salt has been associated with incident hypertension due in part to the increase in extracellular volume [1]. However, recent evidence has shown that apart from renal regulation of sodium homeostasis, other mechanisms are responsible for regulating sodium without commensurate effects on extracellular volume [2, 3]. Blood vessels and red blood cells are lined by negatively charged glycosaminoglycans (GAGs) or glycocalyx which buffer sodium ions (Na+) by binding to them [2, 4]. The negative charges on the GAGs of blood vessels and red blood cells repel each other ensuring that the GAGs are not corroded to expose endothelial lining and therefore cause endothelial activation [2, 4, 5]. Too much sodium could saturate the buffering capacity of GAGs resulting in eroding of the GAGs, endothelial activation and extravasation of sodium into the interstitium to elicit an inflammatory cascade. A "salt blood test" was developed that estimates the buffering capacity of red blood cells and indirectly, that of the GAGs of the blood vessel by reporting erythrocyte sodium sensitivity (ESS) percentage, a value that is inversely proportional to the sodium buffering capacity [5–7]. Individuals who consume large amounts of salt may saturate their GAGs buffering capacity and tend to have higher ESS values reflecting damage to the endothelial layer [8]. A low sodium buffering capacity correlates with significant changes in plasma sodium concentration and higher blood pressure, in this way increasing the risk for the development of hypertension. Data on ESS and its relation to salt consumption is limited in different populations. The aim of the study was to explore the relationship between erythrocyte sodium buffering capacity, sociodemographic and clinical factors, and self-reported salt consumption at Livingstone Central Hospital.

## Methods

### Study design and population

This was a cross sectional study conducted at Livingstone Central Hospital. We recruited 242 health care workers attending to routine checkups at the hospital. This population included permanent health workers, health care workers on internship and medical students.

### Eligibility

We purposively selected all those that volunteered, who were adults aged 18 years and above and had signed a consent form to participate. We excluded participants that were terminally ill and those with suspected neuropathy and suspected thrombosis, where the risk of ischemic injury is worsened by cuff, and unable to remain supine.

### Ethics approval and consent to participate

Ethical approval was obtained from the Mulungushi University School of Medicine and Health Sciences Research Ethics Committee (IRB: 00012281 FWA: 0002888) on 10th December 2020. Permission to conduct the study was granted by Livingstone Central Hospital Administration. Participants signed written consents before they were recruited in the study. The data collected was de-identified and used for research purposes only.

## Data collection

Interviewer structured questionnaires were used for data collection. Participants were interviewed and immediately examined for clinical and laboratory characteristics. ESS was the outcome variable.

ESS was determined by a Salt Blood Test (SBT-mini, Germany) that utilizes capillary blood [6]. The SBT-mini has five components to successfully perform the test: safety lancet, Minivette©, 0.2ml PCR soft tube filled with 50 μl of Na+ cocktail (CARE diagnostica Laborreagenzien GmbH, Voerde, Germany), hematocrit tube Safecap P75-2000M (Scholz GmbH, Neubiberg, Germany) and a plexiglass home-built tube holder. To perform the test, a 50 μl of blood sample was obtained from the fingertip using the EDTA-coated minivette after puncture and the blood was mixed with the Na+ cocktail. The mixed blood was then transferred into a hematocrit tube which was then fixed in a vertical position on a plexiglass home-built tube holder for one hour. The supernatant length was then measured at the end of 60 minutes and ESS calculated. The length of the individual supernatant (in mm) was divided by the mean length of supernatants of the standard (males = 21.4 mm; female = 26.1 mm). This ratio multiplied by 100 expresses the ESS of an individual in percent. 100 ± 20% indicates average ESS of the healthy population [6]. ESS absolute values fall into three categories: <80%, 80–120% and >120% representing low, average and high ESS. A low and average ESS represents good sodium buffering capacity of both the red blood cells and the vessel GAGs while a high ESS means poor sodium buffering capacity.

Apart from the ESS, we also performed ankle brachial index (ABI) as another vascular function characteristic. ABI compares the blood pressure in the ankle with the blood pressure in the arm assisted by a vascular doppler ultrasound. The ABI was performed by measuring the systolic blood pressure from both brachial arteries and from both the dorsalis pedis and posterior tibial arteries after the patient had been at rest in the supine position for 10 minutes. The procedure for ABI is harmless and painless and did not pose any serious risk to the participants. An ABI <0.90 was designated as peripheral artery disease.

Diagnosis of hypertension was based on history of antihypertensive medication usage or clinic BP values of ≥140/90 mmHg on more than 2 occasions.

We used three questions to assess salt consumption behavior of participants on cooking, at the table when eating their meals and their perception on how much salt they consumed, adapted from the validated food frequency questionnaires [9–11]. To estimate salt consumption, the participants were asked how often they added salt while cooking with these structured responses provided for the participants to choose from: always or often times, rarely or never. Participants were also asked how often they added salt while eating on the table with similar structured responses to choose from. Participants were then asked how often they consumed processed and cooked foods high in salt. Tied to this question, the participants were asked to indicate overall, their perception of the amount of salt they were consuming: either right amount to too much (high intake) or very little (low intake). Finally, we randomly sampled a quarter of the total participants who provided 24-hour urine samples for us to estimate dietary salt intake. We used this data to validate responses on perceived salt intake from the questionnaire.

## Data analysis

We exported the data into SPSS software for analysis. We used descriptive statistics to describe our data using median (interquartile range, IQR) for continuous variables and percentage for categorical data. We used Chi-square test with adjusted standard residuals (ASR) to compare proportions between ESS and categorical independent characteristics. We used Kruskal-Wallis

to compare continuous variables among the ESS categories and linear regression analysis to compare absolute ESS values and independent variables. A p-value of less than 0.05 was considered to be statistically significant. Significant ASR and p values are indicated in bold.

## Reporting format

We have used the strengthening the reporting of observational studies in epidemiology (STROBE). See S1 Table for details.

## Results

### General characteristics

Median age of participants was 27 years (22–42, IQR), Table 1. 57% (n = 138) were males. The majority (n = 150, 62%) of participants had an ESS of greater than 120%. The unemployed had the highest proportion of high ESS compared with the employed. A higher proportion of participants who perceived to consume the right to high amounts of salt in processed and cooked foods were in the high ESS category compared to those consuming little or low amounts.

**Table 1. Sociodemographic factors associated with ESS.**

| Variable | | | ESS n (%) | | | P value |
|---|---|---|---|---|---|---|
| | | | Low | Average | High | |
| **Age, median** *years* | | 27 (22, 42) | 31 (22, 47) | 30 (23, 42) | 25 (22, 40) | 0.19 |
| **BMI** *kg/m²* | | | 23.3 (20.0, 27.2) | 23.9 (21.0,28.1) | 24.4 (21.1, 27.8) | 0.87 |
| **Gender,** *n = 242* | *Male* | | 21 (67.7) | 43 (70.5) | 74 (49.3) | **0.008** |
| | *ASR* | | 1.3 | **2.5** | -3.1 | |
| | *Female* | | 10 (32.3) | 18 (29.5) | 76 (50.7) | |
| | *ASR* | | -1.3 | -2.5 | **3.1** | |
| **Employment** | *Employed* | | 11 (35.5) | 23 (37.7) | 30 (20.0) | **0.014** |
| | *ASR* | | 1.2 | 2.3 | -2.9 | |
| | *Unemployed* | | 20 (64.5) | 38 (62.3) | 120 (80.0) | |
| | *ASR* | | -1.2 | -2.9 | **2.9** | |
| **Marital status** | *Married* | | 8 (25.8) | 28 (45.9) | 53 (35.3) | 0.45 |
| | *Single* | | 20 (64.5) | 30 (49.2) | 91 (60.7) | |
| | *Divorced/separated* | | 2 (6.5) | 2 (3.3) | 3(2.0) | |
| | *Widowed* | | 1 (3.2) | 1 (1.6) | 3 (2.0) | |
| **History of smoking** | *Yes* | | 5 (20.0) | 11 (22.0) | 21 (15.1) | 0.50 |
| | *No* | | 20 (80.0) | 39 (78.0) | 118 (84.9) | |
| **Alcohol intake** | *Yes* | | 5 (20.0) | 13 (26.0) | 32 (23.2) | 0.83 |
| | *No* | | 20 (80.0) | 37 (74.0) | 106 (76.8) | |
| **Adds salt on table** | *Always or sometimes* | | 21 (67.7) | 43 (70.5) | 110 (73.3) | 0.13 |
| | *Rarely or never* | | 10 (32.3) | 18 (29.5) | 40 (26.7) | |
| **Adds salt while cooking** | *Always or sometimes* | | 29 (93.5) | 61 (100) | 144 (96.0) | 0.19 |
| | *Rarely or never* | | 2 (6.5) | 0 (0.0) | 6 (4.0) | |
| **Self-reported consumption of salt in processed and cooked food** | *Right amount to too much (high)* | | 11 (35.5) | 27 (44.3) | 96 (64.0) | **0.002** |
| | *ASR* | | -2.4 | -2.0 | **3.4** | |
| | *Little or low* | | 20 (64.5) | 34 (55.7) | 54 (36.9) | |
| | *ASR* | | 2.4 | 2.0 | **-3.4** | |

ESS, erythrocyte sodium sensitivity; ASR, adjusted standardized residual; BMI, body mass index.

**Table 2. Clinical factors associated with ESS.**

| Variable | | ESS n(%) | | | P value |
|---|---|---|---|---|---|
| | | **Low** | **Average** | **High** | |
| **FBS** *mmol/l, n = 193* | | 5.0 (4.7, 5.4) | 4.3 (3.9, 5.0) | 4.4 (4.0, 4.9) | 0.77 |
| **Hypertension status, n = 233** | *Hypertensive* | 2 (6.7) | 6 (10.2) | 10 (6.9) | 0.71 |
| | *Normotensive* | 28 (93.3) | 53 (89.8) | 134 (93.1) | |
| **HIV status**, *n = 242* | *HIV negative* | 29 (93.5) | 55 (90.2) | 135 (90.0) | 0.82 |
| | *HIV positive* | 2 (6.5) | 6 (9.8) | 15 (10.0) | |
| **Antiretroviral therapy class**, *n = 17* | *NNRTI* | 0 (0.0) | 3 (75.0) | 5 (41.7) | 0.43 |
| | *INSTIs* | 1 (100.0) | 1 (25.0) | 4 (33.3) | |
| | *PIs* | 0 (0.0) | 0 (0.0) | 3 (25.0) | |
| **ABI**, *n = 210* | *Normal* | 26 (96.3) | 41 (89.1) | 130 (94.4) | 0.31 |
| | *PAD* | 1 (3.7) | 5 (10.9) | 7 (5.1) | |
| **History of Tuberculosis**, *n = 22* | *Yes* | 2 (100) | 1 (16.7) | 3 (21.4) | 0.05 |
| | *No* | 0 (0.0) | 5 (83.3) | 11 (78.6) | |
| **Diabetes mellitus type 2**, n = 242 | *Yes* | 3 (9.7) | 2 (3.3) | 3 (2.0) | 0.09 |
| | *No* | 28 (90.3) | 59 (96.7) | 147 (98.0) | |
| **Alanine amino transferase**, *mmol/l, n = 37* | | 24 (20, 39) | 61 (44, 101) | 22 (17, 44) | **0.013** |
| **Aspartate amin transferase**, *mmol/l, n = 35* | | 29 (21, 48) | 27 (23, 49) | 23 (19, 28) | 0.16 |
| **Total cholesterol**, *mmol/l, n = 145* | | 3.9 (3.4, 4.5) | 3.9 (3.2, 4.3) | 3.8 (3.2, 4.6) | 0.98 |
| **Hemoglobin**, *g/dl, n = 228* | | 15.0 (13.1, 15.3) | 14.7 (13.1, 15.6) | 13.7 (12.1, 14.7) | **0.001** |
| **Hematocrit**, *%, n = 219* | | 43 (34, 47) | 44 (39, 46) | 40.7 (35.9, 44) | **0.002** |
| **RBC** x $10^{12}$/L, *n = 230* | | 4.9 (4.3, 5.4) | 4.9 (4.5, 5.5) | 4.6 (4.0, 5.0) | **<0.001** |
| **RDW, %,** *n = 229* | | 13.4 (12.8, 14.9) | 13.9 (13.0, 14.8) | 13.7 (12.9, 14.9) | 0.73 |

ESS, erythrocyte sodium sensitivity; ASR, adjusted standardized residual; RDW, red cell distribution width; RBC, red blood cell; FBS, fasting blood sugar; ABI, ankle brachial index; NNRTI, non-nucleoside reverse transcriptase inhibitors; INSTIs, Integrase strand transfer inhibitors; PAD, Peripheral artery disease; PIs, protease inhibitors.

## Clinical characteristics of the participants

The majority were normotensive (92%, n = 215) and HIV negative (90%, n = 219), Table 2. Participants with high ESS values had lower hemoglobin, red blood cells and hematocrit values compared to those with low and average ESS while the level of alanine amino transferase activity was higher for participants with average ESS compared to those with low and high ESS.

## Simple and multiple linear regression results

On simple linear regression Red blood cells (Fig 1A) and hemoglobin (Fig 1B) concentration was negatively associated with ESS values. One unit decrease in red blood cells and hemoglobin was associated with eleven and four unit increases in absolute ESS values, respectively. High self-reported salt consumption (Fig 1C) and estimated salt intake (Fig 1D) were positively associated with higher ESS values.

On multiple linear analysis, we included all variables associated with ESS on simple linear regression (model 1, Table 3) and only hemoglobin and high self-reported salt intake remained significantly associated with ESS. Since hemoglobin is found inside red blood cells, we removed it in model 2 to determine the effect of red blood cells on ESS and found that it remained significantly and negatively associated with ESS. An additional third model where we included estimated salt intake from 24-hour urine samples showed a significant positive

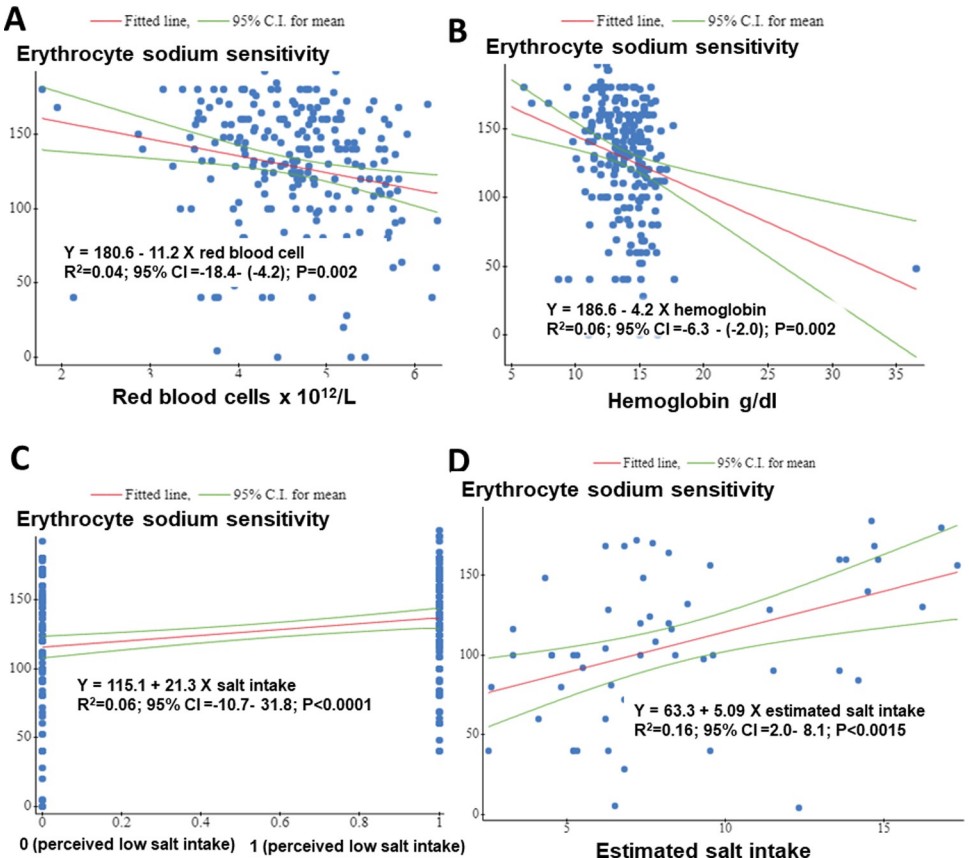

**Fig 1. Relationship between erythrocyte sodium sensitivity (ESS) and covariates in simple linear regression.** ESS absolute values were positively associated with self-reported high salt intake (C), estimated salt intake (D) and negatively associated with red blood cell count (A) and hemoglobin concentration (B). y = erythrocyte sodium sensitivity.

**Table 3. Multiple linear analysis of factors associated with erythrocyte sodium sensitivity.**

| Characteristic | beta | Standard error | p | 95% CI |
|---|---|---|---|---|
| Model 1* | | | | |
| Female | 2.54 | 6.32 | 0.68 | -9.93–15.01 |
| Unemployed | 5.87 | 6.09 | 0.33 | -6.13–17.89 |
| Red blood cell count | -6.76 | 4.13 | 0.10 | -14.91–1.37 |
| Hemoglobin | -3.07 | 1.27 | **0.01** | -5.58–0.55 |
| Self-reported high salt intake | 20.63 | 5.34 | **<0.001** | 10.11–31.16 |
| Model 2** | | | | |
| Female | 8.84 | 6.15 | 0.15 | -3.29–20.97 |
| Unemployed | 7.08 | 6.23 | 0.25 | -5.20–19.36 |
| Red blood cell count | -8.77 | 3.98 | **0.02** | -16.61–0.92 |
| Self-reported high salt intake | 20.55 | 5.47 | **<0.001** | 9.77–31.33 |

*All factors statistically significant were added to the multilinear model

**hemoglobin removed.

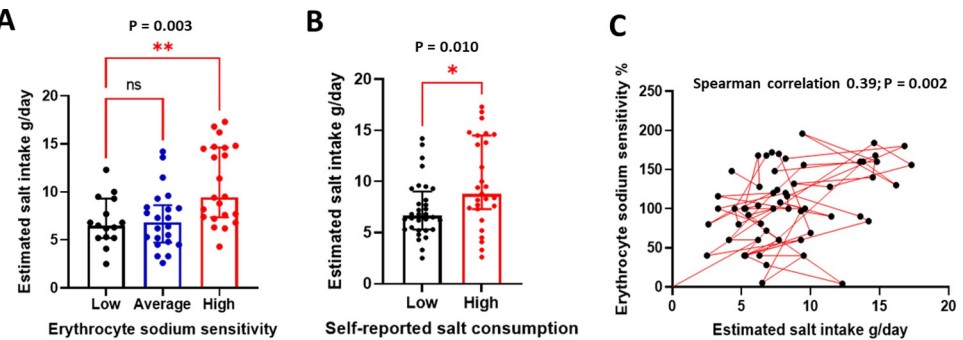

**Fig 2. Estimated salt intake compared to ESS and self-reported salt consumption.** In (A), participants in high ESS category consumed more salt compared to those in the low category. In (B), individuals who reported to consume high salt had higher estimated salt intake compared to those who self-reported to consume low salt. In (C), ESS correlated positively with estimated salt intake from 24-hour urine samples.

association between ESS and actual salt intake (S2 Table). However, none of the variables that were significant in model 1 and 2 remained significant in the third model.

In order to validate data on self-reported salt intake, we sampled 24-hour urine samples to estimate actual dietary salt intake. The average (interquartile range) estimated salt intake in the population was 7.6 (5.5, 10.0) g/day and was incremental on ESS categories as expected [Low 6.5 (5.2, 9.3) vs Average 6.8 (4.7, 8.6) vs High 9.4 (7.3, 14.6) g/day, p = 0.003], Fig 2A. In addition, participants who perceive to consume high salt consumed more salt than those who perceived to consume low salt [Low 6.6 (5.3, 9.0) vs High 8.8 (7.3, 14.5) g/day, p = 0.01], Fig 2B. ESS correlated positively with estimated salt intake, Fig 2C.

## Discussion

The aim of our study was to explore the relationship between erythrocyte sodium buffering capacity, sociodemographic and clinical factors, and self-reported salt consumption at Livingstone Central Hospital. We found that a higher proportion (62%, n = 150) of participants had high ESS compared with participants with average (25%, n = 61) and low (13%, n = 31) ESS. High self-reported salt intake was associated with high ESS corresponding with poor erythrocyte sodium buffering capacity. Negatively charged surfaces of erythrocytes (RBC) reflect properties of the endothelial glycocalyx or GAGs. Therefore, a low buffering capacity of RBCs may likely indicate a damaged glycocalyx.

Our study suggests that self-reported high salt intake is associated with damage to the glycocalyx. Moreover, that the majority of participants had poor sodium buffering capacity was remarkable but plausible with the fact that the majority reported to have high salt intake and the sample came from a population known to be highly salt sensitive [12]. Salt sensitivity of blood pressure (BP) refers to the BP responses for changes in dietary salt intake [13]. Individuals with meaningful BP increases or decreases following increases or decreases in salt intake, respectively, are termed salt sensitive while those who do not exhibit such changes in BP are called salt resistant [14]. Thus, BP in salt sensitive individuals correlates positively with salt intake and ESS [4, 6, 13]. Although we did not compare BP values with ESS, it has been reported that excess salt intake has pathological effects on the buffering capacity and function of the vasculature that are independent of BP [13].

### Relationship between ESS, erythrocyte and the glycocalyx buffering capacity

The glycocalyx is a dense layer of sugars, proteins and lipids that serve several functions in maintaining blood flow and endothelial health [15]. It is mainly composed of sugars and sugar

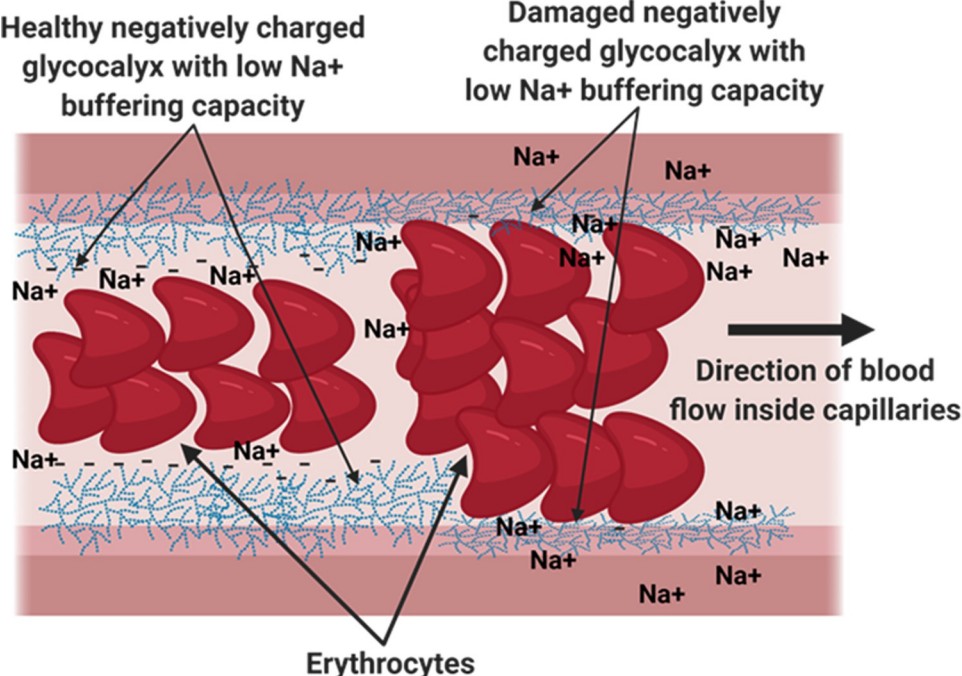

**Fig 3. Illustration depicting the roles of erythrocytes and the glycocalyx in sodium buffering.** The Glycocalyx and erythrocytes are negatively charged and buffer sodium in the vessel. A low buffering capacity leads to friction between the erythrocytes and the glycocalyx resulting in endothelial damage, activation and inflammation which are risk factors for cardiovascular disease.

conjugates such as proteoglycans and glycoproteins, covering the endothelium luminally and separating blood cells, mainly erythrocytes from endothelial cells [16]. The glycocalyx is negatively charged and so are the erythrocytes. The negative charges repel the erythrocyte from the glycocalyx and in this way avoids friction between the two. Sodium (Na+) is positively charged and exerts a rather high affinity to the erythrocyte surface and glycocalyx. The glycocalyx and erythrocytes function to buffer normal plasma levels without saturating their negative charges, however, high sodium levels can reduce the buffering capacity of both and hence lead to friction between the erythrocytes and the glycocalyx (Fig 3).

In a separate study, a low erythrocyte buffering capacity (high ESS) correlated with higher blood pressure [6]. Because red blood cells play an important role in buffering sodium ions [5, 6], it is expected that red blood cell count correlates inversely with ESS and our study confirmed this hypothesis.

## Clinical implications

From a clinical stand-point, given the high proportion of individuals with high ESS, there exists no standard management guidelines to mitigate such findings in patients. Although, there is enough evidence indicating that sodium restrictive diets are beneficial for cardiovascular health and blood pressure regulation [1, 2], dietary salt intake is often overlooked when assessing patients in the clinic. We therefore recommend an inclusion of dietary assessments in routine clinics and sodium restrictive behavior for this population. A high ESS indicates high intake of sodium behavior, which is a risk factor for hypertension. Our findings further confirmed that ESS correlated positively with estimated salt intake from random 24-hour urine samples assayed. Since estimating sodium intake from a 24-hour urine sample is in most

cases not feasible in the clinic, ESS could be considered as a proxy for sodium intake-associated vascular function in future studies. However, this requires validation using already established vascular assessments.

## Limitations

There was no detailed assessment of diet in general, so future studies should explore other dietary factors that may correlate with ESS. The ESS test has not been validated to estimate sodium buffering capacity in our population. More robust techniques such as magnetic resonance imaging and GlycoCheck system are therefore required to validate our findings. This is our goal for future studies.

## Strengths of the study

This study utilizes a cheap, simple and fast capillary blood test (ESS) to assess vascular function related to handling of sodium. This is particularly important in low-cost settings where the cost for vascular assessment is prohibitive and where the burden of cardiovascular disease is likely high. The study also utilized a validated tool for assessing self-reported salt intake which has been reported to correlate significantly with actual salt intake [9–11, 17]. We further estimated dietary salt intake from 24-hour urine samples to ascertain how much salt the participants consumed. Estimated dietary salt consumption correlated positively with self-reported salt consumption. This is similar to a prior study where they found that self-reported salt habit or consumption correlates positively with actual salt intake [18]. Therefore, our data and that of others validates our method of assessing self-reported salt consumption using the questionnaire. We have further showed that intake of salt was higher than that recommended by the World Health Organization.

## Conclusions

Self-reported high salt intake, lower hemoglobin or red blood cell count was associated with poor vascular Na+ buffering capacity in majority of middle-aged Zambians living in Livingstone. Self-reported salt intake correlated positively with actual salt consumption estimated for 24-hour urine samples. The results of our study provide the potential for ESS to be considered as a proxy for dietary salt-intake-associated vascular function in clinical practice. However, future studies need to validate vascular function status when using ESS testing by including established vascular function assessments to determine the pathophysiological and clinical implications of the ESS test. To obtain a more accurate measure of sodium intake, future studies should use urinary biomarker methods to estimate dietary intake of salt and also explore the factors associated with high ESS in a hypertensive Zambian population. Sodium restrictive diets are recommended to improve vascular health in this population.

## Supporting information

**S1 Table. Strobe checklist.**
(DOCX)

**S2 Table. Supplementary data.**
(DOCX)

**S1 Data.**
(XLSX)

## Acknowledgments

We want to thank Mulungushi University medical students: Robert Musekwa, Shimumba Mesha, Joy Hangoma, Carol Macha, Valerie Hayumbu and Fiona Sitali who are part of HAND RG research for their support during data collection. We also thank the Mulungushi University Management for supporting this study.

## Author Contributions

**Conceptualization:** Sepiso K. Masenga.

**Data curation:** Sepiso K. Masenga, Malani Malumani, Benson M. Hamooya.

**Formal analysis:** Sepiso K. Masenga, Benson M. Hamooya.

**Investigation:** Sepiso K. Masenga.

**Methodology:** Sepiso K. Masenga.

**Project administration:** Sepiso K. Masenga, Benson M. Hamooya.

**Resources:** Sepiso K. Masenga.

**Software:** Sepiso K. Masenga.

**Supervision:** Sepiso K. Masenga.

**Validation:** Sepiso K. Masenga, Leta Pilic, Malani Malumani.

**Visualization:** Sepiso K. Masenga, Benson M. Hamooya.

**Writing – original draft:** Sepiso K. Masenga, Leta Pilic, Malani Malumani, Benson M. Hamooya.

**Writing – review & editing:** Sepiso K. Masenga, Leta Pilic, Malani Malumani, Benson M. Hamooya.

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
