## [Decision Letter · Decision Letter 0]

22 Oct 2021

PONE-D-21-29014Erythrocyte sodium buffering capacity status correlates with self-reported salt intake in a population from Livingstone, ZambiaPLOS ONE

Dear Dr. Masenga,

Thank you for submitting your manuscript to PLOS ONE. After careful consideration, we feel that it has merit but does not fully meet PLOS ONE’s publication criteria as it currently stands. Therefore, we invite you to submit a revised version of the manuscript that addresses the points raised during the review process.

Two expert reviewers have carefully studied your manuscript. Fortunately, both find the topic of interest. However, a number of important concerns are raised that must all be addressed.A more accurate measure of salt consumption would considerably increase the relevance of your findings.

We look forward to receiving your revised manuscript.

Kind regards,

Jaap A. Joles, DVM, PhD

Academic Editor

PLOS ONE

Journal Requirements:

2. Please include additional information regarding the questionnaire used in the study and ensure that you have provided sufficient details that others could replicate the analyses.

Reviewers' comments:

Reviewer's Responses to Questions

**Comments to the Author**

1. Is the manuscript technically sound, and do the data support the conclusions?

Reviewer #1: Partly

Reviewer #2: Yes

2. Has the statistical analysis been performed appropriately and rigorously? 

Reviewer #1: Yes

Reviewer #2: Yes

3. Have the authors made all data underlying the findings in their manuscript fully available?

Reviewer #1: Yes

Reviewer #2: No

4. Is the manuscript presented in an intelligible fashion and written in standard English?

Reviewer #1: No

Reviewer #2: No

5. Review Comments to the Author

Reviewer #1: Masenga and colleagues want to define what factors are related to sodium buffering capacity in a population living in Zambia. To establish sodium buffering capacity they have used the erythrocyte sodium sensitivity test as developed by Oberleithner a couple of year ago. The idea that sodium can be stored in the endothelial glycocalyx and the glycocalyx of the erythrocyte may have great impact on the current concepts of sodium homeostasis and pathophysiology of salt sensitivity. As such, this paper addresses an important topic, but I have some important concerns.

Major comments

- Abstract

o In the introduction the authors state that high sodium consumption leads to reduced sodium buffering capacity. I don’t think this statement is completely true. Data from animals and intervention studies indicate that high sodium effects can be neutralized by increasing buffering capacity, for instance, in the skin (Machnik A, Nature). Glycocalyx dimension might change after an IV sodium load, but necessarily after high sodium intake (Rorije NM, Anesthesiology)

o Results. Line 39. The word ‘about’ should be removed. Please report exact numbers in the result section.

o Results. Line 41. The expression ‘erythrocyte sodium sensitive’ is an unclear expression and seems more an interpretation of the results than a true finding.

- Methods section

o The description of the included patients is very unclear. It seems a pretty random selection of health care workers and patients visiting the hospital on a regular base.

o The way sodium consumption has been estimated is not clearly described. For instance, what does an adapted version of the food frequency questionnaire exactly mean?

o The ESS test that has been used for estimating sodium buffering capacity has in my view never been validated. Sodium buffering capacity can be demonstrated after tissue ashing methods or by MRI techniques. Unfortunately, no data with regard to these accepted methods are available to my understanding.

- Results

o Line 124. Please remove 'about'.

o Line 125 reads a bit difficult.

o Table 2 shows that red blood cell indices (RBC, Hb, and Ht) differ for each ESS category. Given the lack of validation of the ESS test, I wonder whether ESS is not merely another red blood cell marker than a reflection sodium buffering in glycocalyx. It seems that the results of the multiple linear analysis do not take away this concern.

o Figure 1 A and B show the regression between categorical variables. This is not very informative.

Reviewer #2: Masenga performed a very interesting study showing a correlation between the erythrocyte glycocalyx, ABI and self-reported salt consumption in a healthy cohort.

This is of great importance and can be useful as low-cost, fast and simple tool for the prevention of hypertension and cardiovascular events which are often triggered by high salt consumption, especially in salt-sensitive individuals.

However, I have some concerns:

1. A clear description of the methods is missing and should be provided

2. Also, a clear structure of the manuscript is missing (e.g. figure legends appear in the discussion section)

3. A self-reported salt consumption is a limitation. At least random sampling of 24 hour urine should be done

4. At least an estimation of the salt consumption should be given (e.g. 1 tea spoon)

5. What is the authors explanation of the fact that unemployed had the highest proportion of high ESS compared with the employed?

6. PLOS authors have the option to publish the peer review history of their article (what does this mean?). If published, this will include your full peer review and any attached files.

Reviewer #1: **Yes: **Liffert Vogt

Reviewer #2: **Yes: **Kristina Kusche-Vihrog

---

## [Author Response · Author response to Decision Letter 0]

7 Jan 2022

1st December, 2021

To the Editor in Chief

Dear Editor,

Ref: Re-submission of revised research article

The above matter refers. 

We thank the reviewers for their constructive and positive review of our manuscript. We revised the manuscript in accordance with the various review comments (indicated in bold) and hope that it is now acceptable for publication. I am pleased to resubmit a revised version of our manuscript. Details of the changes are shown in the “track changes” version of the manuscript. 

Responses to the reviewers

Reviewer #1: Masenga and colleagues want to define what factors are related to sodium buffering capacity in a population living in Zambia. To establish sodium buffering capacity, they have used the erythrocyte sodium sensitivity test as developed by Oberleithner a couple of year ago. The idea that sodium can be stored in the endothelial glycocalyx and the glycocalyx of the erythrocyte may have great impact on the current concepts of sodium homeostasis and pathophysiology of salt sensitivity. As such, this paper addresses an important topic, but I have some important concerns.

Major comments

- Abstract

In the introduction the authors state that high sodium consumption leads to reduced sodium buffering capacity. I don’t think this statement is completely true. Data from animals and intervention studies indicate that high sodium effects can be neutralized by increasing buffering capacity, for instance, in the skin (Machnik A, Nature). Glycocalyx dimension might change after an IV sodium load, but necessarily after high sodium intake (Rorije NM, Anesthesiology)

Response: We agree with the reviewer that this statement is misleading. We have corrected it to imply only that there is a possible relationship between salt intake and sodium buffering capacity and CVD which is supported by literature. We appreciate this comment.

Results. Line 39. The word ‘about’ should be removed. Please report exact numbers in the result section.

Response: We have removed as suggested. Thank you

Results. Line 41. The expression ‘erythrocyte sodium sensitive’ is an unclear expression and seems more an interpretation of the results than a true finding.

Response: Thank you for the suggestion. We have modified appropriately and reported only the finding.

Methods section

The description of the included patients is very unclear. It seems a pretty random selection of health care workers and patients visiting the hospital on a regular base.

Response: We agree very much with the reviewer that the population under study was not described clearly. We have clarified this by adding some descriptive details. The population consisted of health care workers attending to routine medical checkups. We excluded any participants who were ill at the time of study.

The way sodium consumption has been estimated is not clearly described. For instance, what does an adapted version of the food frequency questionnaire exactly mean?

Response: We have now elaborated in detail how salt consumption was assessed in the structured questionnaire. In addition, as requested by another reviewer, we have additionally sampled a quarter of the total participants and estimated their dietary salt intake using 24-hour urine samples to validate perceived salt intake. We found a positive correlation and our data is further supported by prior research. 

The ESS test that has been used for estimating sodium buffering capacity has in my view never been validated. Sodium buffering capacity can be demonstrated after tissue ashing methods or by MRI techniques. Unfortunately, no data with regard to these accepted methods are available to my understanding.

Response: We agree that ESS is not validated to estimate sodium buffering capacity in our population and we have therefore been very cautious on our interpretation of data. We have added this limitation in the limitation section of the discussion.

Results

Line 124. Please remove 'about'.

Response: We have removed ‘about’ as suggested. Thank you.

Line 125 reads a bit difficult.

Response: We have re-written this sentence to read clearly. Thank you.

Table 2 shows that red blood cell indices (RBC, Hb, and Ht) differ for each ESS category. Given the lack of validation of the ESS test, I wonder whether ESS is not merely another red blood cell marker than a reflection sodium buffering in glycocalyx. It seems that the results of the multiple linear analysis do not take away this concern.

Response: Much appreciated for this suggestion. Our understanding of ESS test is that since the buffering capacity of red blood cells mirror that of the glycocalyx, a reduction in RBC count would mean a reduction in the buffering of sodium. This is expected as shown in the linear regression results and table 2. But truly, this does not take away the possibility that ESS may just be another red blood cell marker, a concern raised by the reviewer. However, we have now provided supplementary data (S2 table) that was based on additional analysis where we have included estimated salt intake in the multiple linear analysis and this takes away the concern of the red cell indices; they did not remain significantly associated with ESS when actual estimated salt intake was included in the model. Thank you for this suggestion.

Figure 1 A and B show the regression between categorical variables. This is not very informative.

Response: Thank you for the suggestion. We have removed the two figures on gender nd employment

Reviewer #2: Masenga performed a very interesting study showing a correlation between the erythrocyte glycocalyx, ABI and self-reported salt consumption in a healthy cohort.

This is of great importance and can be useful as low-cost, fast and simple tool for the prevention of hypertension and cardiovascular events which are often triggered by high salt consumption, especially in salt-sensitive individuals.

However, I have some concerns:

1. A clear description of the methods is missing and should be provided

Response: We have expanded the methods section to accurately elaborate the methods to ensure that they are reproducible. We appreciate the concern and fill that the methods section is now clearly described. We are happy to receive further specific advice if necessary.

2. Also, a clear structure of the manuscript is missing (e.g. figure legends appear in the discussion section)

Response: This is very true for most journals. However, we noticed that journal requirement for plos one do not prohibit this. We have adhered to the specific journal requirements. It appears that it is a journal requirement to add figure legends immediately after the figure caption and title. We thank you for this concern.

3. A self-reported salt consumption is a limitation. At least random sampling of 24-hour urine should be done

Response: We have managed to perform random sampling of 24-hour urine to estimate actual salt intake and have included this in our results. See Fig 2 also.

4. At least an estimation of the salt consumption should be given (e.g. 1 tea spoon)

Response: We have now provided information on salt consumption in g/day based on 24-hour urine samples. This suggestion is appreciated.

5. What is the authors explanation of the fact that unemployed had the highest proportion of high ESS compared with the employed?

Response: This finding was interesting and would be important to explain further as suggested here. However, It was beyond the scope of our study to explain this finding that the unemployed had the highest proportion of high ESS compared with the employed. Moreover, since this finding was not significant on multilinear regression. We thought it not necessary to discuss further as this would divert focus on the main findings. We hope to look at this more in detail in future studies focused on sociodemographic factors.

We sincerely thank you again for the suggestions which have remarkably improved our manuscript. 

We look forward to hearing from you at your earliest convenience. 

Yours sincerely,

Dr. Sepiso K. Masenga

---

## [Decision Letter · Decision Letter 1]

15 Feb 2022

Erythrocyte sodium buffering capacity status correlates with self-reported salt intake in a population from Livingstone, Zambia

PONE-D-21-29014R1

Dear Dr. Masenga,

We’re pleased to inform you that your manuscript has been judged scientifically suitable for publication and will be formally accepted for publication once it meets all outstanding technical requirements.

Kind regards,

Jaap A. Joles, DVM, PhD

Academic Editor

PLOS ONE

Additional Editor Comments (optional):

The reviewers comments have been adequately addressed.

Unfortunately Reviewer #1 did not respond to my invitation but I am satisfied with your response.

Reviewers' comments:

Reviewer's Responses to Questions

**Comments to the Author**

1. If the authors have adequately addressed your comments raised in a previous round of review and you feel that this manuscript is now acceptable for publication, you may indicate that here to bypass the “Comments to the Author” section, enter your conflict of interest statement in the “Confidential to Editor” section, and submit your "Accept" recommendation.

Reviewer #2: All comments have been addressed

2. Is the manuscript technically sound, and do the data support the conclusions?

Reviewer #2: Yes

3. Has the statistical analysis been performed appropriately and rigorously? 

Reviewer #2: Yes

4. Have the authors made all data underlying the findings in their manuscript fully available?

Reviewer #2: Yes

5. Is the manuscript presented in an intelligible fashion and written in standard English?

Reviewer #2: Yes

6. Review Comments to the Author

Reviewer #2: After expanding the methods section and the addition of information about the cohort and salt consumption, my concerns are addressed.

7. PLOS authors have the option to publish the peer review history of their article (what does this mean?). If published, this will include your full peer review and any attached files.

Reviewer #2: **Yes: **Kristina Kusche-Vihrog

---

## [Editor Report · Acceptance letter]

21 Feb 2022

PONE-D-21-29014R1 

Erythrocyte sodium buffering capacity status correlates with self-reported salt intake in a population from Livingstone, Zambia 

Dear Dr. Masenga:

I'm pleased to inform you that your manuscript has been deemed suitable for publication in PLOS ONE. Congratulations! Your manuscript is now with our production department. 

Kind regards, 

on behalf of

Dr. Jaap A. Joles 

Academic Editor

PLOS ONE